# Proteomic Analysis of Irradiation with Millimeter Waves on Soybean Growth under Flooding Conditions

**DOI:** 10.3390/ijms21020486

**Published:** 2020-01-12

**Authors:** Zhuoheng Zhong, Takashi Furuya, Kimitaka Ueno, Hisateru Yamaguchi, Keisuke Hitachi, Kunihiro Tsuchida, Masahiko Tani, Jingkui Tian, Setsuko Komatsu

**Affiliations:** 1Faculty of Environment and Information Sciences, Fukui University of Technology, Fukui 910-8505, Japan; zhongzhh@zju.edu.cn (Z.Z.); shizouliu@gmail.com (K.U.); 2College of Biomedical Engineering & Instrument Science, Zhejiang University, Hangzhou 310027, China; tjk@zju.edu.cn; 3Research Center for Development of Far-Infrared Region, University of Fukui, Fukui 910-8507, Japan; furuya@fir.u-fukui.ac.jp (T.F.); tani@fir.u-fukui.ac.jp (M.T.); 4Institute for Comprehensive Medical Science, Fujita Health University, Toyoake 470-1192, Japan; hyama@fujita-hu.ac.jp (H.Y.); hkeisuke@fujita-hu.ac.jp (K.H.); tsuchida@fujita-hu.ac.jp (K.T.)

**Keywords:** early-stage soybean, seed irradiation, crop productivity, glycolysis, redox

## Abstract

Improving soybean growth and tolerance under environmental stress is crucial for sustainable development. Millimeter waves are a radio-frequency band with a wavelength range of 1–10 mm that has dynamic effects on organisms. To investigate the potential effects of millimeter-waves irradiation on soybean seedlings, morphological and proteomic analyses were performed. Millimeter-waves irradiation improved the growth of roots/hypocotyl and the tolerance of soybean to flooding stress. Proteomic analysis indicated that the irradiated soybean seedlings recovered under oxidative stress during growth, whereas proteins related to glycolysis and ascorbate/glutathione metabolism were not affected. Immunoblot analysis confirmed the promotive effect of millimeter waves to glycolysis- and redox-related pathways under flooding conditions. Sugar metabolism was suppressed under flooding in unirradiated soybean seedlings, whereas it was activated in the irradiated ones, especially trehalose synthesis. These results suggest that millimeter-waves irradiation on soybean seeds promotes the recovery of soybean seedlings under oxidative stress, which positively regulates soybean growth through the regulation of glycolysis and redox related pathways.

## 1. Introduction

The Leguminosae comprise about 18,000 species, most of which are able to participate in nitrogen-fixing symbioses [1]. These plants play an important ecological role as interesting sources of vegetable proteins [2]. Soybean milk has a considerably high protein content, which matches the value of milk from cows [3]. The transition of protein supply from animals to plants is a desirable strategy to keep up with the increasing protein demand caused by accelerating prosperity and world population increases [4]. Soybean is an important crop for both protein meal and vegetable oil [5]. Unfortunately, the growth and yield of a soybean are strongly affected by diverse environmental conditions [6]. These findings indicate that improvement of soybean quality and tolerance under environmental stress is utmostly crucial.

Many crops, including soybean, are particularly sensitive to stress from flooding [7,8], which causes an increase of the carbon dioxide concentration in soil [9] and inhibits development of underground tissues [10]. Furthermore, ATP production and energy transformation through oxidative phosphorylation are impaired, which severely affect soybean growth [11]. Lateral root and root hairs begin to grow shortly after the germination of radicles within five days of planting [12]. During these five days, the flooding injury to seedlings before germination which was caused by a physical disruption of the rapid uptake of water could be alleviated by using seeds with high moisture contents [13]; however, the injury after germination induced cell death of the root tip containing the root apical meristem and severely threated the root system development of soybean [14,15], which resulted in suppressed length/weight of underground tissues [16]. To promote plant growth and development, several strategies were used to improve early-stage soybean tolerance under flooding conditions. Soybean plants supplied with abscisic acid had a high survival ratio after flooding compared to control [17]. Flood-tolerant soybeans, which were isolated from gamma-ray irradiated mutants, also recovered after water removal [18]. Flooded soybean recovered after water removal following treatment with plant-derived smoke [19]. These applications have significant influence towards the development of a flood-tolerant soybean, however, they are not environmentally friendly. Investigations of simple and safe applications are needed to protect soybean plants from flooding damage.

Millimeter waves are a radio frequency band with a range of 30–300 GHz, which corresponds to a wavelength range of 1–10 mm [20]. Millimeter-waves irradiation is an environmentally appropriate technology, with long wavelengths and small threats to human health [21], which has been reported to have dynamic effects on organisms. It influences the vital activity, cell division, enzyme synthesis, and biomass accumulation of several microorganisms [22]. Wheat seeds which were treated with the millimeter waves, grew into higher plants with better biological harvest yields, compared to those without treatment [23]. On the contrary, millimeter-waves irradiation had no positive impact on potato growth [24]. Therefore, the millimeter-waves irradiation may be an effective approach to promote harvesting in some crops, however, the effects on soybeans have not been investigated.

Pre-sowing treatment of the seeds has advantages such as simplicity and economic feasibility. Pre-sowing treatment of soybean seeds with magnetic field [25], laser [26], or nanoparticles [27] affected soybean growth and resulted in better accumulation of the nutritional compounds, regardless of the rigid seed coat. Treatment of soybean seeds with millimeter-waves irradiation might be a potential method for enhancing soybean growth and development, which may positively regulate soybean metabolism under stressful conditions; however, the effects of millimeter waves on soybean have not been studied. To determine the effects of the millimeter-waves irradiation on soybean growth, morphological analysis was performed through comparisons among treatments in different ranges of dose and duration of the irradiation. Based on the morphological results, proteomic analysis was carried out to explore the responsible mechanisms for the positive effects of millimeter waves on soybean growth and flooding recovery. Proteomic results were subsequently confirmed by immunoblot analysis.

## 2. Results

### 2.1. Morphological Analysis of Soybean Irradiated with Millimeter Waves

To investigate the effect of millimeter-waves irradiation on soybean and soybean subjected to flooding stress, the morphological changes of soybean after different treatments were analyzed (Appendix A). Soybean seeds were irradiated with 5, 10, 20, and 40 mW of millimeter waves, while untreated seeds were used as control. The growth of soybean seedlings was suppressed by flooding (Figure 1). Millimeter-waves irradiation promoted the elongation of hypocotyl of soybean and soybean with flooding stress. The length and weight of hypocotyl of soybean, which treated with 10 mW irradiation, were the highest compared to those treated with other doses of irradiation (Appendix A). To further investigate the optimal time for irradiation, soybean seeds were irradiated or not with 10 mW of millimeter waves for 10, 20, 40, and 80 min. The duration for 20 min was most effective for promoting the growth of hypocotyl and root (Figure 1). Based on the morphological results, millimeter-waves irradiation of 10 mW for 20 min was set as the pre-sowing treatment for further proteomic analysis.

### 2.2. Identification and Functional Investigation of Proteins in Root-Hypocotyl Tissue of Soybean Irradiated with Millimeter Waves

In order to explore the effect on growth of soybean seeds irradiated with millimeter waves, a gel-free/label-free proteomic technique was used. Proteins were extracted from the root-hypocotyl tissue of soybean before (starting point) and after treatment. There were four groups of soybeans: irradiated/unirradiated and flooded/unflooded. The relative abundance of peptides and proteins from irradiated soybean was compared with that from unirradiated soybean. The proteomic data of all samples from different groups were compared by principle component analysis (PCA), which indicated the different expression patterns of proteins from different treatment (Appendix A). In total, the abundance of 1151, 674, and 1919 proteins differentially changed in millimeter-waves irradiated soybean at the starting point, unflooded, and flooded conditions, respectively, compared to soybean without irradiation (Appendix A). Among them, 66 proteins were commonly identified from three comparisons. The functional category of these proteins was determined using MapMan bin codes. Commonly changed proteins were mainly involved in protein synthesis/targeting, photosynthesis, development, and cell division (Table 1). Proteins related to hormone metabolism such as 12-oxophytodienoate reductase and lipoxygenase increased in irradiated soybeans at the starting point, unflooded and flooded conditions compared to unirradiated ones.

Furthermore, proteins related to photosynthesis significantly increased in irradiated soybeans from these three comparisons, and the fold change of photosystem I reaction center subunit III and photosystem I P700 chlorophyll a apoprotein A2 were 3.00 and 3.04 under flooded conditions compared to unirradiated soybeans. Especially, chaperonin 10 increased with a fold change of 5.47 in irradiated soybean seedlings compared to unirradiated ones at the starting point. To further investigate the effect of millimeter-waves irradiation during soybean growth under unflooded and flooded conditions, relative abundance of peptides and proteins from unflooded and flooded conditions was compared with that from the starting point. Under unflooded condition, there are 2227 and 697 proteins differentially changed in unirradiated and irradiated soybean seedlings, respectively, during growth. (Appendix A). Under flooded condition, those numbers were 1903 and 1473, respectively, before and after flooding stress (Appendix A). The functional category of these proteins was determined using MapMan bin codes (Figure 2). Within unirradiated soybeans, altered proteins during under both unflooded and flooded conditions were mainly involved in protein synthesis/degradation/targeting, RNA regulation, cell, and stress (as the percentages in totally altered proteins under unflooded/flooded conditions were 18.6%/20.2%, 16.3%/6.7%, 4.7%/4.3%, and 4.6%/5.0%, respectively). Within irradiated soybeans, percentages of proteins related to these four categories were also high, however, proteins related to other categories also intensively changed. Namely, 4.2% (29/697) of altered proteins under unflooded conditions related to cell wall; 4.5% (67/1473) of altered proteins under flooded conditions related to amino acid metabolism, while only 2.0% (14/697) of proteins related to this category altered under unflooded conditions.

### 2.3. Identification of Metabolic Pathways Related to Millimeter-Waves Irradiation in Soybean during Growth

To visualize the effect of millimeter-waves irradiation to metabolic pathways during soybean growth, proteins with differential abundance were submitted to MapMan software and mapped (Figure 3). Under unflooded conditions, proteins in irradiated soybeans related to Calvin cycle, ascorbate/glutathione cycle, and glycolysis were less affected (Figure 3A,B); the numbers of altered proteins related to these three pathways were 1, 5, 4, respectively, accounted for 0.6% (1/162), 3.0% (5/162), and 2.5% (4/162). In the meanwhile, those numbers in unirradiated soybeans were 15, 18, and 23, accounted for 3.2% (15/473), 3.8% (18/473), 4.9% (23/473), accordingly. Furthermore, under flooded conditions, proteins in unirradiated soybean related to photorespiration, nucleotide metabolism, and oxidative pentose phosphate pathway decreased; the percentage of decreased ones to altered proteins within each category was 100% (6/6), 58% (19/33), and 80% (4/5), respectively (Figure 3C). Proteins in irradiated soybean related to photorespiration, nucleotide metabolism and oxidative pentose phosphate were less inhibited compared to the unirradiated group; as the corresponding percentage of decreased proteins was 40% (2/5), 36% (9/25), 30% (3/10) (Figure 3D). In addition, the abundance of pyruvate kinase related to glycolysis increased in control group during flooding, whereas that in millimeter waves irradiated group decreased during flooding instead (Figure 3C,D), which implied the differential regulation on glycolysis of millimeter-wave irradiation under flooding.

As proteins related to glycolysis and ascorbate/glutathione cycle were differentially changed in 4-day old seedlings compared to 2-day old seedlings (Figure 3), which may differently regulate energy metabolism and plant resistance during growth [28,29]. To visualize the effect of millimeter-waves irradiation to soybean growth, changed proteins related to glycolysis or ascorbate/glutathione metabolism were mapped based on KEGG database (Figure 4 and Figure 5).

In the glycolysis pathway, without millimeter-wave treatment, proteins were significantly affected during unflooded growth; namely, nine out of 15 changed proteins were decreased (Figure 4A). In irradiated soybeans, only two out of five changed proteins were decreased during unflooded growth (Figure 4B). Nevertheless, flooding induced the upregulation of glycolysis within both unirradiated and irradiated seedlings as important regulation for energy production under hypoxia conditions [30], from which more than half of the proteins increased in both irradiated and unirradiated soybean seedlings (Figure 4C,D). In ascorbate/glutathione pathway, under unflooded conditions, the abundance of ascorbate peroxidase increased in unirradiated plants and the abundance of glutathione dehydrogenase decreased; however, these two proteins did not change in irradiated plants (Figure 5A,B). Under flooded conditions, ascorbate peroxidase decreased in unirradiated soybean seedlings and increased in irradiated ones (Figure 5C,D), which presented as the only protein that was differentially expressed in response to flooding.

Considering the fact that oxygen is deprived under flooding stress, which hinders aerobic respiration, sugar metabolism correlating with glycolysis is responsible for energy production during flooding [31]. Differentially expressed proteins were mapped to sugar metabolism based on KEGG database. Under unflooded condition, many of related proteins decreased in unirradiated soybeans while only fructokinase decreased in irradiated soybeans and other proteins increased or did not change (Figure 6A,B). In response to flooding, sugar metabolism is up-regulated, proteins such as hexosephosphate aminotransferase, UDP glucose pyrophosphorylase, UDP-glucose dehydrogenase, and glucurono kinase increased (Figure 6A,C). Furthermore, millimeter-wave irradiation promoted the up-regulation of sugar metabolism. As indicated, proteins such as chitinase, hexokinase, 4-α-glucanotransferase, sucrose-phosphate synthase, and phospho-glucomutas decreased in unirradiated soybean seedlings during flooding (Figure 6C); however, these proteins increased in irradiated soybean seedlings (Figure 6D), which were responsible for the supply of precursors for energy production [28].

### 2.4. Immunoblot Analysis of Proteins involved in Ascorbate/Glutathione Metabolism and Glycolysis in Soybean Root during Flooding

To better uncover the change of proteins from different treatments, immunoblot analysis of proteins related to ascorbate/glutathione metabolism and glycolysis were performed. Proteins were extracted from root-hypocotyl tissue of soybeans which treated with varied doses (0, 5, 10, and 20 mW) of millimeter waves under unflooded or flooded condition. CBB staining pattern was used as loading control (Appendix A). To confirm the finding according to proteomic result, the presence of APX, GR in ascorbate and glutathione metabolism were analyzed; To analyze the expression of PRX, which was essential for redox signaling and flooding response in soybean [32], anti-PRX antibodies was used (Appendix A). Under unflooded condition, APX and PRX increased in response to 10mW of millimeter waves, while GR did not change. In response to flooding stress, the expression patterns in unirradiated and 10 mW-millimeter-irradiated soybeans were same- APX decreased and PRX increased; however, the abundance of both APX and PRX were higher in soybeans irradiated with 10 mW millimeter-waves, compared to unirradiated ones (Figure 7). For investigation of change in glycolysis, the presence of FBPA, TPI, and GAPDH were analyzed (Appendix A). FBPA is not affected by neither flooding stress nor millimeter-waves irradiation. TPI and GAPDH in soybean increased in response to flooding stress. In the meanwhile, the abundance of TPI and GAPDH in soybeans that irradiated with 10 mW-millimeter waves were higher than that in unirradiated soybeans under flooded condition (Figure 8), which suggested that the millimeter-waves irradiation promoted the up-regulation of glycolysis. 

## 3. Discussion

### 3.1. Millimeter-Waves Irradiation on Soybean Seeds Improved Early-Stage Soybean Growth and Flooding Tolerance

The electromagnetic field of the millimeter waves band, which is used in satellite communication, radiometry, radar, and remote sensing technologies, has emerged as a novel and widespread environmental factor [33]. Millimeter-waves irradiation promotes the growth of plants via increase of seedling length [34,35], leaf weight [36], shoot multiplication [37], or advance of pollen-tube emergence [38]. In the present study, the effect of millimeter-waves irradiation on the early-stage development of soybean seeds was analyzed. Morphological analysis indicated that the millimeter-waves irradiation not only improved the development of root and hypocotyl in 4-day old soybean plants, but also promoted soybean tolerance during flooding stress (Figure 1). Altered proteins at 2-day stage, 4-day-without-flooding stage, and 4-day-with-flooding stage were investigated. Proteins related to hormone metabolism and photosynthesis increased in irradiated soybeans at all three stages, which suggests that essential growth-regulating metabolisms are activated during early-stage growth. Most importantly, chaperonin 10 increased in irradiated soybean seedlings compared to unirradiated ones at the starting point (Table 1). Chaperonin 10 plays a prominent role in plastid protein folding especially the assembly of ribulose bisphosphate carboxylase oxygenase [39], which had protective effect on plants by establishing normal protein conformation and cellular homeostasis [40]. In soybean, reduced expression of chaperonins resulted in abnormal mitochondria metabolism under chilling temperatures [41]. Present results with previous findings revealed the regulatory role of millimeter-waves irradiation on plant growth.

Despite of the countless evidence for the promotive effect of millimeter-waves irradiation on plant growth, few studies investigated its effect under abiotic stress. Flooding represents a severe abiotic stress which threats the growth and yield of many crops [6,8]. Flooding injury during early-stage development post germination induces cell death of the root tip [14] and largely inhibited lateral root development [15]. Therefore, increase of soybean flooding tolerance at this stage is a large promise for improving soybean long-term development and productivity. Several strategies have been applied to increase soybean tolerance under flooding [17,18,19], yet a simple and safe strategy is needed. Present results suggest that millimeter-waves irradiation had promotive effect on soybean growth under flooding stress, where elongation of root and hypocotyl was recovered after water removal. Our findings demonstrated the importance of millimeter-waves irradiation to soybean root growth. The improved root growth guaranteed soybean recovery after flooding stress, which increased the possibility of soybean surviving from waterlog and contributed to the higher productivity.

### 3.2. Millimeter-Waves Irradiation Promoted Carbohydrate Metabolism for Energy Production

Germination is a turning point when plants convert from anaerobic-dominant respiration to aerobic-dominant respiration [17]. Generation of reactive-oxygen species increases after germination, which inhibits multiple glycolytic enzymes [42]. In the present study, the abundance of GAPDH, enolase, and pyruvate kinase decreased during growth in unirradiated soybeans, which is consistent with previous knowledge; however, these proteins did not change during growth in irradiated soybeans. Despite these changes, the abundance of phospho-hexokinase both decreased in unirradiated and irradiated soybeans. (Figure 4). Phospho-hexokinase catalyzes the irreversible conversion of glucose to glucose-6-phosphate, which is the rate-limiting enzyme in glycolysis [43]. Immunoblot analysis revealed that there were no significant changes among the soybeans treated with varied doses of irradiation under unflooded condition (Figure 8). Current results indicated that millimeter-waves irradiation promoted glycolysis in the soybean, but the regulatory effect might be inhibited by the up-stream rate-limiting step.

Flooding inhibited the oxidative phosphorylation of ADP into ATP in the soybean roots [44], impaired carbohydrate metabolism [28], and resulted in the dominance of substrate-level of phosphorylation in ADP to ATP (which utilizes glucose or fructose via glycolytic-fermentation pathway [45]). Sugar metabolism is responsible for the catalyzation of glycolytic precursors such as glucose and its phosphates, and it was pointed out as the tolerant-responsive process at the initial flooding stress in a soybean [46], where gene expression levels of phosphoglucomutase and sucrose-phosphate synthase are upregulated in flooding-tolerant mutants compared to wild-type plants [33]. Therefore, alteration of sugar metabolism altogether with its down-stream glycolysis is an important adaptation of plants under flooding stress [45]. The species or genotype with greater carbohydrate concentration in roots, and an efficient metabolic mechanism associated with carbohydrate mobilization and fermentation pathway, would be more suitable to face oxygen deprivation under flooding [47]. In the present study, proteins which are related to sugar metabolism from unirradiated and irradiated soybean seedlings both, intensively changed under flooding stress (Figure 6). However, in unirradiated group, more than half of the altered proteins decreased, whereas those proteins increased in millimeter waves irradiated group. Chitinase, which catalyzes the hydrolytic cleavage of chitin to GlcNAC [48], was considered as a defense mechanism against pathogens and abiotic stress [49]. Glycogen phosphorylase was responsible for the phosphorolytic degradation of starch into glucose-1P [50]; glucanotransferase catalyzed the transfer of maltooligosaccharides from maltose to glucose [51], both provided precursors for glycolysis. Immunoblot analysis indicated that the abundances of TPI and GAPDH in irradiated soybeans were higher than those in unirradiated soybeans under flooded condition (Figure 8). GAPDH reversibly catalyzes the formation of glyceraldehyde to 1,3-bisphosphoglycerate, which was increased under various stress such as anaerobic [52], salt [53], and flooding [31]. Moreover, although GAPDH is not a rate-limiting enzyme in glycolysis, increased GAPDH mediated the glycolysis rate as a mechanism by which the plant cell prepared for a demand of ATP [29]. These findings suggest that millimeter-waves irradiation alters sugar metabolism of soybean during flooding stress, which might promote the accumulation of soluble sugars and positively facilitate soybean growth through upregulation of GAPDH in glycolysis.

Interestingly, proteins which are related to trehalose synthesis such as trehalose-6P synthase and trehalose-phosphatase increased in irradiated soybean seedling during flooding stress (Figure 6). Trehalose functions as a storage carbohydrate and protects individuals against a variety of stress in bacteria, fungi, and insects; while, in plants these roles are largely replaced by sucrose [54]. Nevertheless, several studies indicated that trehalose and trehalose-6P are essential for embryogenic and vegetative growth of *Arabidopsis thaliana* [55,56,57]. Present results with previous knowledge suggest that trehalose synthesis is activated in soybean seedlings irradiated with millimeter waves under flooding stress, which may alternatively promote the plant growth and tolerance of soybean under flooding stress.

### 3.3. Millimeter-Waves Irradiation Differentially Regulated Redox-Related Pathways for Tolerance under Oxidative Stress

Metabolism of ascorbate and glutathione forms the main detoxification system of active oxygen species operating in plant cells [58]. The ascorbate-glutathione cycle not only combats oxidative stress, but also plays an important role in various developmental processes [59,60]. In the present study, the change patterns and the presence of GR in unirradiated soybeans or irradiated soybeans were both similar under unflooded and flooded conditions (Figure 5 and Figure 7); However, APX underwent differential changes. APX decreased in response to flooding in unirradiated soybeans but increased in irradiated soybeans (Figure 5). Immunoblot analysis indicated that APX increased in soybeans which irradiated with optimum power strength compared to that in unirradiated soybeans (Figure 7), which supported that millimeter-waves treatment induced the expression of APX. Ascorbate peroxidase catalyzed the transformation of ascorbate and H_2_O_2_ into H_2_O and dehydro-ascorbate, which is important for the scavenging of oxygen species [61]. Proteomic results indicated that the abundance of APX in 4-day old irradiated seedlings did not change compared to 2-day-old irradiated seedlings; the abundance of APX in 4-day-old unirradiated seedlings was higher than 2-day-old unirradiated seedlings. Immunoblot analysis indicated that the abundance of APX in 4-day-old irradiated seedlings was higher than that in 4-day-old unirradiated seedling. Based on these results, although the expression of APX did not change in irradiated soybean seedlings from 2-day-stage to 4-day-stage, it was both higher at 2-day-old and 4-day-old compared to unirradiated seedlings. The high expression level of APX during 2-day-stage to 4-day-stage may promote the soybean growth, considering the fact that soybean is suffering from increasingly oxidative environment after soybean germination during this period [17].

Peroxiredoxins are a ubiquitous class of cysteine-dependent enzymes that degrade hydroperoxides into water [62]. Although catalytic cycle of PRX is excluded from ascorbate and glutathione cycle [63], PRX is important not only as intracellular antioxidants but also as direct participants in redox signaling [64]. According to the immunoblot result, PRX significantly increased in response to flooding stress, which is consistent with previous findings [32]. Under flooding stress, the accumulation of PRX in irradiated soybeans were higher than that in unirradiated soybeans (Figure 7). PRX from grapes responded to environmental signals such as heat, light, and pathogens [65]. Furthermore, PRX from grapes was proved to be involved with H_2_O_2_ signaling and abscisic-acid signaling in the abiotic stress protection [30]. Similar regulatory mechanisms may occur in a soybean under abiotic stress, and the increase of PRX induced by millimeter waves may promote soybean tolerance under flooding.

## 4. Materials and Methods

### 4.1. Plant Material, Millimeter Wave Irradiation, and Treatment

For millimeter wave irradiation, a Gunn oscillator (J.E. Caristrom Co., Chicago, IL, USA) was used as a light source. The Gunn oscillator delivers an output power of 7–80 mW in a frequency range of 79–115 GHz. The Gunn oscillator operated free run at a frequency 110 GHz and output power adjusted by attenuator. The millimeter wave is radiated to the free space by a horn antenna at an angle of 17 degrees. The seeds of soybean (*Glycine max* L. cultivar Enrei) were spread on a Petri dish. For the irradiation intensity of millimeter waves, irradiation was performed for 10 min of 0, 5, 10, 20, and 40 mW, corresponding to average intensities of 0, 0.06, 0.13, 0.25, and 0.51 mW/cm^2^, respectively. For the response to the irradiation time, 10 mW of millimeter-wave power was irradiated for 0, 10, 20, 40, and 80 min (Appendix A). The temperature rise of soybeans was estimated to be well below 1 K, even with the maximum irradiation intensity.

After irradiation, seeds were sterilized with 3% sodium hypochlorite solution, rinsed twice in water, and sown in 400 mL of silica sand in a seedling case. A total of 14 seeds were sown evenly in each seedling case depend on following experiments. Soybeans were grown at 25 °C and 60% humidity under white fluorescent light (160 µmol m^−2^·s^−1^, 16 h light period/day). To induce flooding stress, water was added above the soil surface to immerse 2-day-old soybeans after sowing for 2 days. For flooding recovery, water was removed at 4 days after sowing, and 4-day-old soybeans were moved to normal growing conditions for additional 4 days. For morphological analysis, root and hypocotyl of 8-day-old soybeans were collected with 10 plants for each point. For proteomic analysis, root-hypocotyl tissue of 2-day-old and 4-day-old soybeans were collected with five plants for each point. Three independent experiments were performed as biological replications for all experiments, meaning that the seeds were sown on different days. Irradiated/unirradiated and flooded/unflooded soybeans were collected (Appendix A).

### 4.2. Protein Extraction

A portion (300 mg) of root-hypocotyl tissue was cut into small pieces and put into a filter cartridge. It was ground with a plastic rod 120 times in 75 µL of lysis buffer, which contains 7 M urea, 2 M thiourea, 5% CHAPS, and 2 mM tributylphosphine. The suspension was incubated for 2 min at 25 °C and centrifuged twice with 15,000× *g* at 4 °C for 5 min. The detergents from the supernatant were removed using the Pierce Detergent Removal Spin Column (Pierce Biotechnology, Rockford, IL, USA). The Bradford method [66] was used to determine the protein concentration with bovine serum albumin used as the standard.

### 4.3. Protein Enrichment, Reduction, Alkylation, and Digestion

Extracted proteins (100 µg) were adjusted to a final volume of 100 µL. Methanol (400 µL) was added to each sample and mixed before addition of 100 µL of chloroform and 300 µL of water. After mixing and centrifugation at 20,000× *g* for 10 min to achieve phase separation, the upper phase was discarded and 300 µL of methanol was added to the lower phase, and then centrifuged at 20,000× *g* for 10 min. The pellet was collected as the soluble fraction [18]. Proteins were resuspended in 50 mM ammonium bicarbonate, reduced with 50 mM dithiothreitol for 30 min at 56 °C in the dark, and alkylated with 50 mM iodoacetamide for 30 min at 37 °C in the dark. Alkylated proteins were digested with trypsin and lysyl endopeptidase (Wako, Osaka, Japan) at a 1:100 enzyme/protein ratio for 16 h at 37 °C. Peptides were desalted with MonoSpin C18 Column (GL Sciences, Tokyo, Japan) and acidified with 1% trifluoroacetic acid.

### 4.4. Protein Identification Using Nano-Liquid Chromatography Mass Spectrometry 

The LC conditions as well as the MS acquisition conditions are described in the previous study [19]. The peptides were loaded onto the LC system (EASY-nLC 1000; Thermo Fisher Scientific, San Jose, CA, USA) equipped with a trap column (Acclaim PepMap 100 C18 LC column, 3 µm, 75 µm ID × 20 mm; Thermo Fisher Scientific, San Jose, CA, USA), equilibrated with 0.1% formic acid, and eluted with a linear acetonitrile gradient (0–35%) in 0.1% formic acid at a flow rate of 300 nL min^−1^. The eluted peptides were loaded and separated on the column (EASY-Spray C18 LC column, 3 µm, 75 µm ID × 150 mm; Thermo Fisher Scientific, San Jose, CA, USA) with a spray voltage of 2 kV (Ion Transfer Tube temperature: 275 °C). The peptide ions were detected using MS (Orbitrap Fusion ETD MS; Thermo Fisher Scientific, San Jose, CA, USA) in the data-dependent acquisition mode with the installed Xcalibur software (version 4.0; Thermo Fisher Scientific, San Jose, CA, USA). Full-scan mass spectra were acquired in the MS over 375–1500 m/z with resolution of 120,000. The most intense precursor ions were selected for collision-induced fragmentation in the linear ion trap at normalized collision energy of 35%. Dynamic exclusion was employed within 60 s to prevent repetitive selection of peptides.

### 4.5. Mass Spectrometry Data Analysis

The MS/MS searches were carried out using MASCOT (Version 2.6.1, Matrix Science, London, UK) and SEQUEST HT search algorithms against the UniprotKB *Glycine* max (soybean) protein database (25 October 2017) using Proteome Discoverer (PD) 2.2 (Version 2.2.0.388; ThermoFisher Scientific). The workflow for both algorithms included spectrum files RC, spectrum selector, MASCOT, SEQUEST HT search nodes, percolator, ptmRS, and minor feature detector nodes. Oxidation of methionine were set as a variable modification and carbamidomethylation of cysteine was set as a fixed modification. Mass tolerances in MS and MS/MS were set at 10 ppm and 0.6 Da, respectively. Trypsin was specified as protease and a maximum of two missed cleavage was allowed. Target-decoy database searches used for calculation of false discovery rate (FDR) and for peptide identification FDR was set at 1%.

### 4.6. Differential Analysis of Proteins Using Mass Spectrometry Data

Label-free quantification was also performed with PD 2.2 using precursor ions quantifier nodes. For differential analysis of the relative abundance of peptides and proteins between samples, the freely software PERSEUS (version 1.6.5.0) [67] was used. Proteins and peptides abundances were transferred into log2 scale. Three biological replicates of each sample were grouped and a minimum of three valid values were required in at least one group. Normalization of the abundances was performed to subtract the median of each sample. Missing values were imputed based on a normal distribution (width = 0.3, down-shift = 1.8). Significance was assessed using Student’s *t*-test analysis. PCA was performed with PD 2.2.

### 4.7. Functional Predictions 

Protein functions were categorized using MapMan bin codes [68] and protein abundance ratio was visualized through MapMan software [69] (version 3.6.0RC1; http://mapman.gabipd.org). Pathway mapping of identified proteins was performed using Kyoto Encyclopedia of Genes and Genomes (KEGG) database [70] (http://www.genome.jp/kegg/), in which each square indicates one protein. Red and green squares indicate increased and decreased proteins during growth, respectively.

### 4.8. Immunoblot Analysis

SDS-sample buffer consisting of 60 mM Tris-HCl (pH 6.8), 2% SDS, 10% glycerol, and 5% 2-mercaptoethanol was added to protein samples [71]. Quantified proteins (10 μg) were separated by electrophoresis on a 10% SDS-polyacrylamide gel and transferred onto a polyvinylidene difluoride membrane using a semidry transfer blotter (Nippon Eido, Tokyo, Japan). The blotted membrane was blocked for 5 min in Bullet Blocking One regent (Nacalai Tesque, Kyoto, Japan). After blocking, the membrane was cross-reacted with a 1:1000 dilution of the primary antibodies for 1 h at room temperature. As primary antibodies, the followings were used: anti-ascorbate peroxidases (APX) [72], anti-glutathione reductase (GR) (Agrisera, Vännäs, Sweden), anti-peroxiredoxin (PRX) [44], anti-fructose-bisphosphate aldolase (FBPA) [73], anti-triose phosphate isomerase (TPI) (provided by Proteomic Laboratory in Fukui University of Technology, Fukui, Japan), anti-glyceraldehyde-3-phosphate dehydrogenase (GAPDH) (provided by Proteomic Laboratory in Fukui University of Technology) antibodies. Anti-rabbit IgG conjugated with horseradish peroxidase (Bio-Rad, Hercules, CA, USA) was used as the secondary antibody. After 1 h incubation, signals were detected using TMB Membrane Peroxidase Substrate kit (Seracare, Milford, MA, USA) following the manufacturer’s protocol. Coomassie brilliant blue (CBB) staining was used as loading control. The integrated densities of bands were calculated using Image J software (version 1.8, National Institutes of Health, Bethesda, MD, USA).

### 4.9. Statistical Analysis

The statistical significance of two groups was evaluated by the Student’s *t*-test. The statistical significance of multiple groups was evaluated by one-way ANOVA test. SPSS 20.0 (IBM, Chicago, IL, USA) statistical software was used for the evaluation of the results. A *p*-value of less than 0.05 was considered as statistically significant. 

### 4.10. Accession Code

For MS data, RAW data, peak lists and result files have been deposited in the ProteomeXchange Consortium [74] via the jPOST [75] partner repository under data-set identifiers PXD013889.

## 5. Conclusions

Millimeter-waves irradiation has been proved to act as a growth regulator in several plants [25,26,27,28], however, its effect on soybean has not been studied yet. The productivity of soybean is largely dependent on the plant’s early stage development, during which the soybean growth is easily influenced by environmental stressors such as flooding [6,7]. To investigate the possibility of millimeter-waves irradiation promoting soybean productivity, the potential promotive effect of millimeter-waves irradiation on soybean growth were performed based on morphological, proteomic, and immunoblot analysis. The main findings are: (i) millimeter-waves irradiation of 10 mW (0.13 mW/cm^2^) for 20 min improved the growth of roots/hypocotyl and the tolerance of soybean to flooding stress; (ii) the inhibitory effect to glycolysis during soybean growth recovered as a result of millimeter-waves irradiation; (iii) the soybean seedlings irradiated with millimeter waves were more tolerant to oxidative stress with the up-regulation of APX; and (iv) sugar metabolism was suppressed under flooding in unirradiated soybean seedlings, but it was activated in irradiated soybean seedlings. Taken together, these results suggest that millimeter-waves irradiation has positive effects on soybean seedlings through the regulation of glycolysis and redox related pathways. Furthermore, activation of sugar metabolism in irradiated-soybean seedlings might play an important role in soybean tolerance under flooding stress.

## Figures and Tables

**Figure 1 ijms-21-00486-f001:**
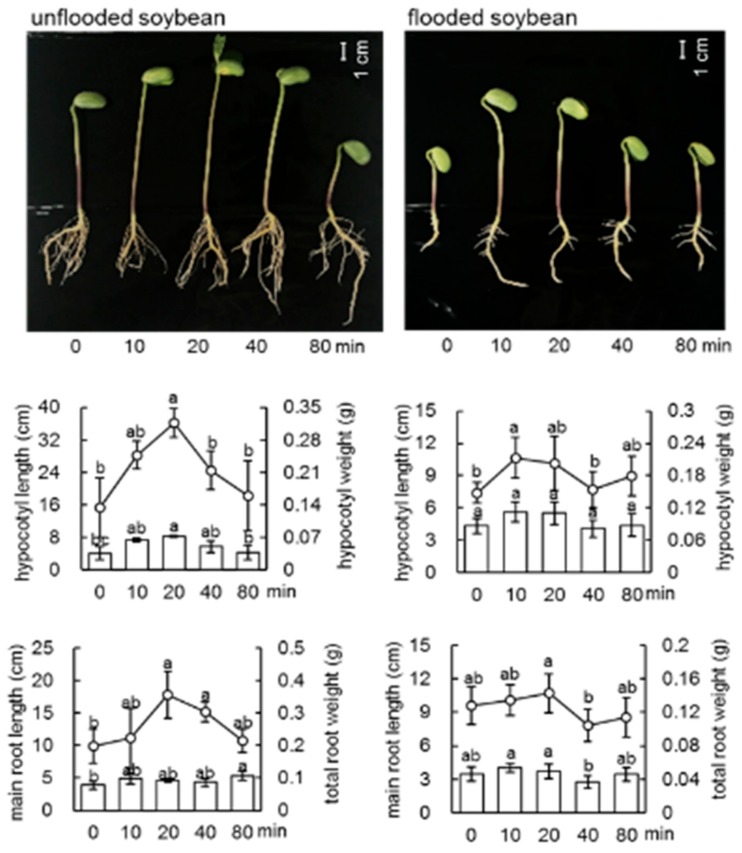
Morphological effect of irradiation of millimeter waves on soybean. Soybean seeds irradiated with millimeter waves were sowed and treated, which is indicated in materials and method. Bar indicates 1 cm. Column graph shows the length of main root/hypocotyl and line graph shows the weight of main root/hypocotyl. Data are shown as means ± SD from three independent biological replicates. The different letters indicate significant changes according to one-way ANOVA followed by Tukey’s multiple comparisons (*p* < 0.05).

**Figure 2 ijms-21-00486-f002:**
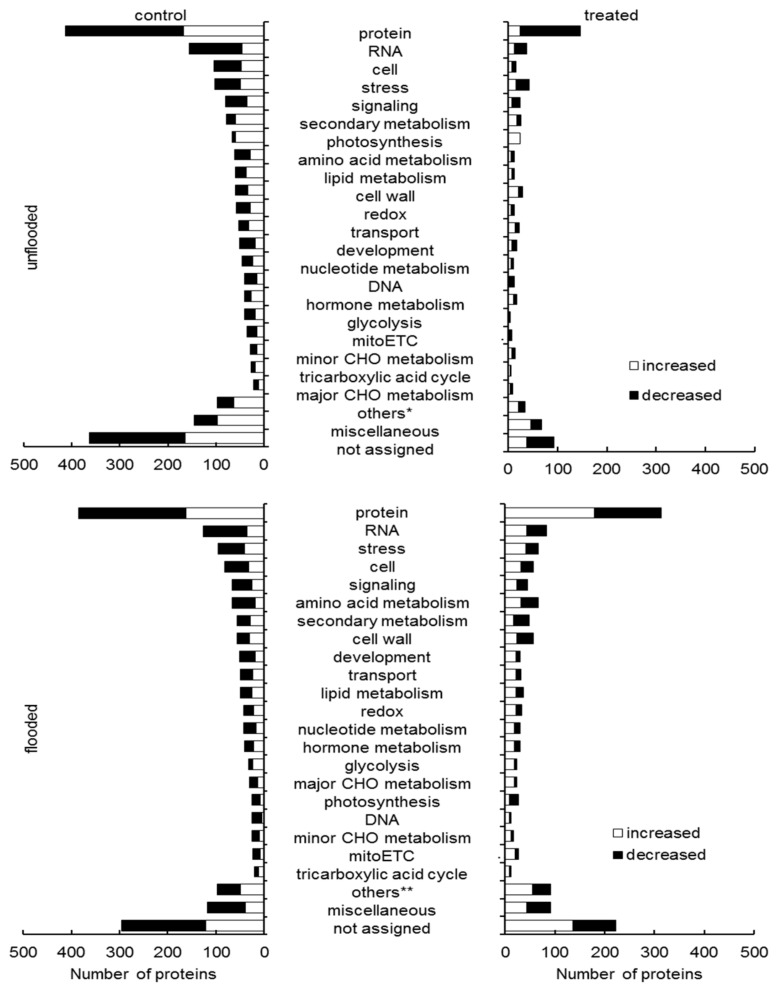
Functional categories of proteins in root-hypocotyl tissue with differential abundance in soybean treated with millimeter-waves irradiation and flooding. Soybean seeds irradiated with millimeter waves were sowed and treated, which is indicated in materials and method. Functional categories of significantly changed proteins (*p* < 0.05) from unirradiated/irradiated and unflooded/flooded during growth were determined using MapMan bin codes. Abbreviation: mitoETC, mitochondrial electron transport chain. “others*” contains proteins related to gluco-neogenesis/glyoxylate cycle, polyamine metabolism, S-assimilation, biodegradation of xenobiotics, N-metabolism, tetrapyrrole synthesis, C1-metabolism, Co-factor and vitamin metabolism, metal handling, and oxidative pentose phosphate pathway. “others**” contains proteins related to gluco-neogenesis, S-assimilation, transporter, polyamine metabolism, biodegradation of xenobiotics, tetrapyrrole synthesis, Co-factor and vitamin metabolism, oxidative pentose phosphate pathway, C1-metabolism, N-metabolism, metal handling, and fermentation. “not assigned” indicates proteins without ontology or characterized functions.

**Figure 3 ijms-21-00486-f003:**
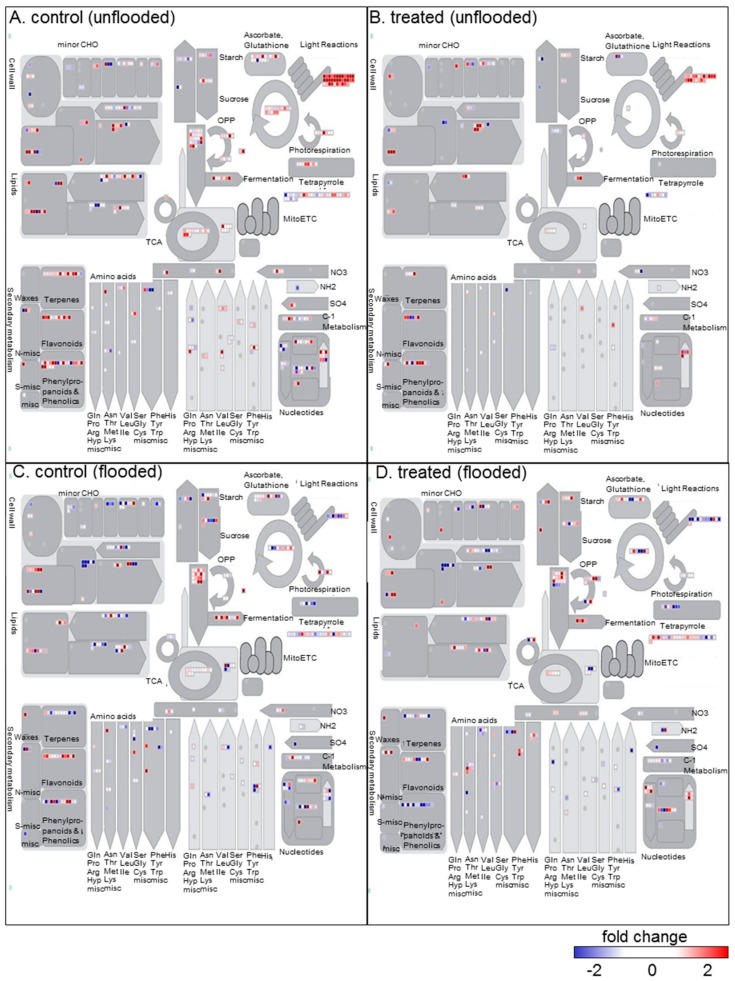
Mapping on metabolic pathways of differentially changed proteins identified from soybean seedling irradiated without (control) or with millimeter waves (treated) under flooded or unflooded conditions. (**A**) Mapping from unirradiated soybean under unflooded conditions. (**B**) Mapping from irradiated soybean under unflooded conditions. (**C**) Mapping from unirradiated soybean under flooding. (**D**) Mapping from irradiated soybean under flooding. Changed proteins were submitted to the MapMan software and mapped to metabolic pathways. Each square indicates one mapped protein. Color indicates the fold change value of a differentially changed protein. Red and blue colors indicate an increase and decrease in fold change values, respectively. Abbreviations: CHO, carbohydrate metabolism; MitoETC, mitochondrial electron transport chain; misc, miscellaneous; OPP, oxidative pentose phosphate pathway; TCA, tricarboxylic acid cycle.

**Figure 4 ijms-21-00486-f004:**
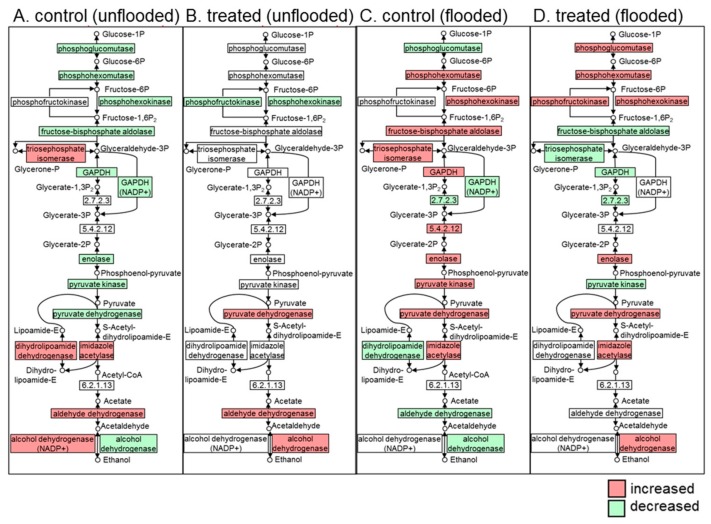
Mapping of differentially changed proteins that related to glycolysis identified from root-hypocotyl tissue of irradiated (treated)/unirradiated soybean (control) during early seedling growth. Proteins were extracted from flooded and unflooded soybean seedlings. (**A**) Mapping from unirradiated soybean under unflooded conditions. (**B**) Mapping from irradiated soybean under unflooded conditions. (**C**) Mapping from unirradiated soybean under flooding. (**D**) Mapping from irradiated soybean under flooding. Proteins neither detected nor changed are marked in white. Abbreviations: GAPDH, glyceraldehyde 3-phosphate dehydrogenase. The number in the box indicates the enzyme commission number of corresponding proteins.

**Figure 5 ijms-21-00486-f005:**
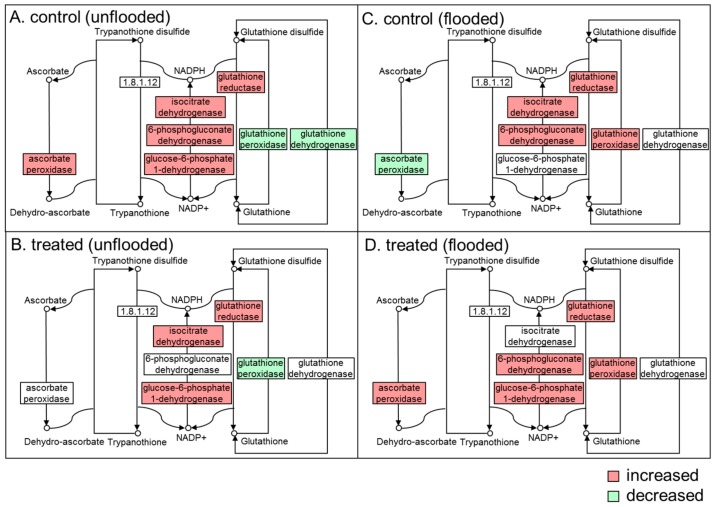
Mapping of differentially changed proteins that related to ascorbate and glutathione metabolism identified from irradiated (treated)/unirradiated (control) soybean root. Proteins were extracted from flooded and unflooded soybean seedlings. (**A**) Mapping from unirradiated soybean under unflooded conditions. (**B**) Mapping from irradiated soybean under unflooded conditions. (**C**) Mapping from unirradiated soybean under flooding. (**D**) Mapping from irradiated soybean under flooding. Proteins neither detected nor changed are marked in white. The number in the box indicates the enzyme commission number of corresponding proteins.

**Figure 6 ijms-21-00486-f006:**
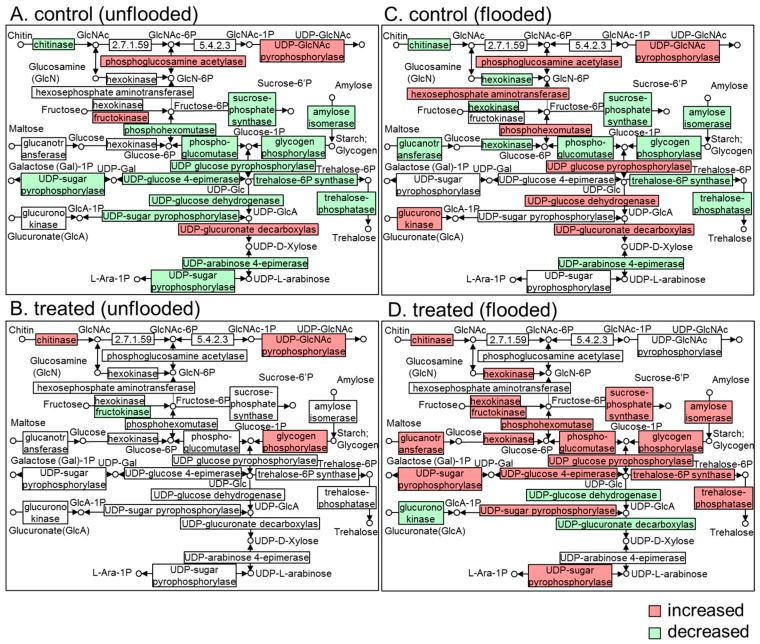
Mapping of differentially changed proteins in soybean root related to sugar metabolism. Proteins were identified from soybean seedlings treated without or with millimeter-waves irradiation under flooded and unflooded conditions. Pathway map was determined on KEGG database. (**A**) Mapping from unirradiated soybean under unflooded conditions. (**B**) Mapping from irradiated soybean under unflooded conditions. (**C**) Mapping from unirradiated soybean under flooding. (**D**) Mapping from irradiated soybean under flooding. Proteins neither detected nor changed are marked in white. The number in the box indicates the enzyme commission number of corresponding proteins.

**Figure 7 ijms-21-00486-f007:**
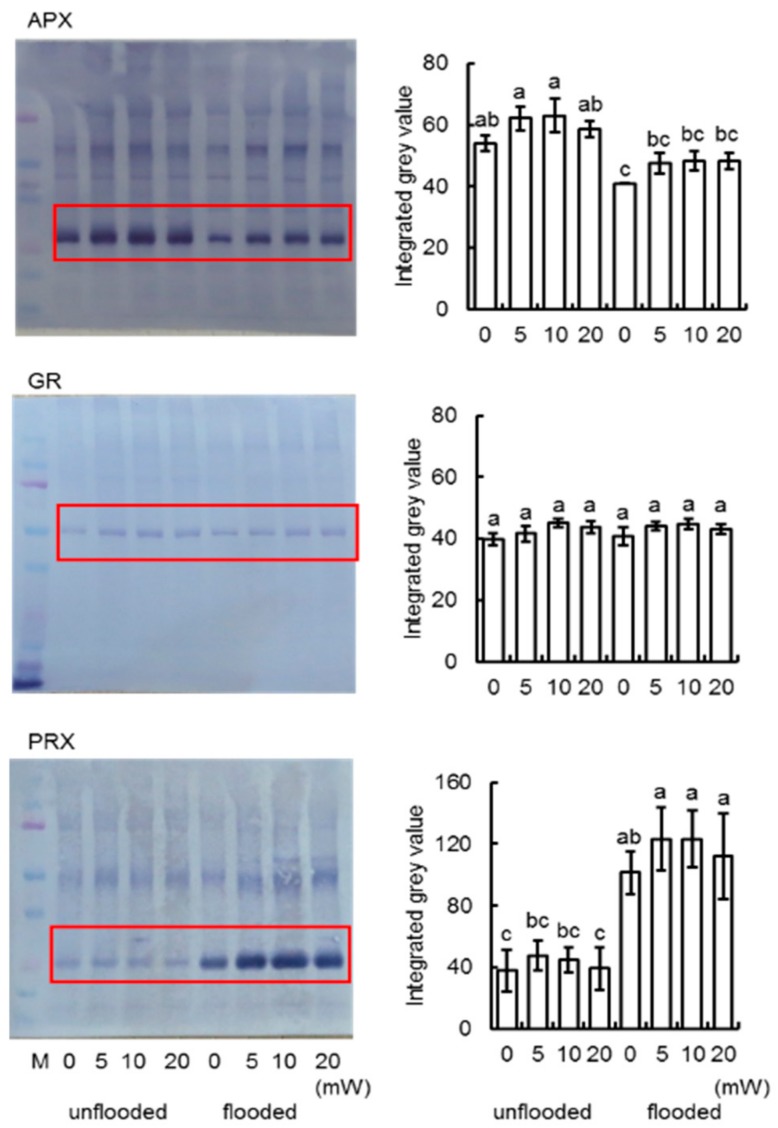
Immunoblot analysis of proteins involved in ascorbate/glutathione pathway. Proteins were extracted from soybean seedlings and separated on 10% SDS-polyacrylamide gel by electrophoresis and transferred onto membranes. The membranes were cross-reacted with anti-APX, anti-GR, and anti-PRX antibodies. CBB staining pattern was used as loading control (Appendix A). The integrated densities of bands were calculated using ImageJ software. Picture shows three independent biological replicates (Appendix A). Data are shown as the means ± SD from three independent biological replicates.

**Figure 8 ijms-21-00486-f008:**
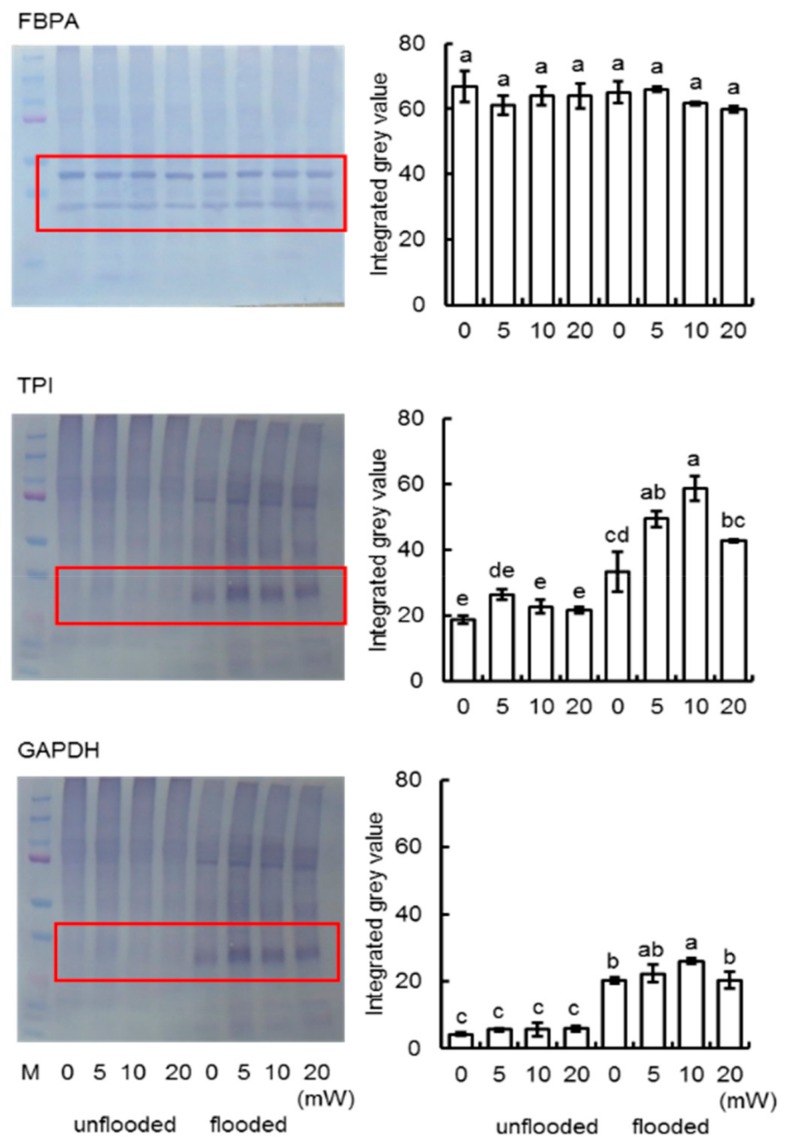
Immunoblot analysis of proteins involved in glycolysis pathway. Proteins were extracted and separated on 10% SDS-polyacrylamide gel by electrophoresis and transferred onto membranes. The membranes were cross-reacted with anti-FBPA, anti-TPI, and anti-GAPDH antibodies. CBB staining pattern were used as loading control (Appendix A). The integrated densities of bands were calculated using ImageJ software. Pictures shows three independent biological replicates (Appendix A). Data are shown as means ± SD from three independent biological replicates.

**Table 1 ijms-21-00486-t001:** List of proteins altered at all three stages (2-day old, 4-day old without flooding, 4-day old with flooding) of soybean irradiated with millimeter waves.

Number	Accession ^a^	Description	Functional Category ^b^	Starting Point	Fold Change ^c^	Flooded
					Unflooded	
1	I1JGU8	phospho-2-dehydro-3-deoxyheptonate aldolase	amino acid metabolism	1.22	−0.61	−0.63
2	I1JEV7	carboxylesterase 2	biodegradation of xenobiotics	0.72	0.35	0.56
3	C6SX10	mitochondrial fission 1 protein	cell	−2.26	−1.39	−1.66
4	Q9M7N4	MFP1 attachment factor 1	cell	−1.51	−2.43	2.60
5	I1N3E1	phosphorylethanolamine cytidylyltransferase 1	cell	−0.72	2.31	−2.30
6	I1K6P4	uncharacterized protein	cell	1.02	−0.66	−1.58
7	Q09WE7	UDP-sugar pyrophosphorylase 1	cell wall	−1.22	0.81	0.38
8	V6CKR0	expansin	cell wall	−0.76	−0.38	0.54
9	Q9SEK9	seed maturation protein PM25	development	−1.75	−2.49	−1.06
10	I1L849	seed maturation protein	development	−0.88	−0.72	−2.80
11	I1LE33	cupin family protein	development	−1.89	−1.83	−2.36
12	I1L860	cupin family protein	development	−3.01	−3.54	−3.28
13	I1KY39	citrate synthase	gluconeogenese/glyoxylate cycle	−0.46	−0.40	0.56
14	I1LY51	12-oxophytodienoate reductase	hormone metabolism	0.70	0.53	0.37
15	A0A0R0IYE6	lipoxygenase	hormone metabolism	0.85	1.43	1.34
16	A0A0R0K553	uncharacterized protein	hormone metabolism	−1.08	−0.67	−0.38
17	A0A0R0GAV8	acyl-CoA dehydrogenase family member 10	lipid metabolism	−1.23	−0.54	0.88
18	I1MRK1	3-hydroxybutyryl-CoA epimerase	lipid metabolism	0.97	0.77	−2.38
19	I1M928	acyl-lipid (9-3)-desaturase isoform B	lipid metabolism	−2.52	1.61	0.98
20	O22378	metallothionein-II protein	metal handling	−3.07	2.56	0.40
21	I1K146	cytochrome c1-1, heme protein	mitoETC	−0.80	−0.60	0.45
22	C6SWW6	cytochrome c oxidase subunit 6a	mitoETC	1.11	−0.35	−0.98
23	I1KEH3	electron transfer flavoprotein subunit alpha	mitoETC	−0.38	−0.32	0.88
24	I1J9Q7	glutamate dehydrogenase	nitrogen metabolism	−1.31	−0.61	3.52
25	I1ML46	uncharacterized protein	nucleotide metabolism	1.53	1.08	1.06
26	Q2PMU2	photosystem I P700 chlorophyll a apoprotein A2	photosynthesis	1.52	0.70	3.04
27	P49161	cytochrome f	photosynthesis	1.21	0.85	0.69
28	A0A0R0I8Z5	chlorophyll a-b binding protein	photosynthesis	1.18	0.91	1.72
29	Q2PMQ9	photosystem II CP47 reaction center protein	photosynthesis	1.12	0.74	1.42
30	A0A0R4J389	photosystem I reaction center subunit III	photosynthesis	1.13	0.94	3.00
31	I1MUQ0	ATP synthase delta chain	photosynthesis	0.73	0.69	1.07
32	K7KJ72	uncharacterized protein	protein. degradation	0.24	−0.38	−0.51
33	I1M4G0	carboxypeptidase	protein. degradation	1.16	0.52	0.81
34	I1MIC1	metacaspase-4	protein. degradation	1.19	−0.29	0.30
35	C6TFY7	chaperonin 10	protein. folding	5.47	−0.63	−6.07
36	I1K5E6	guanine nucleotide-binding protein subunit	protein. post translational modification	0.38	−0.30	−0.48
37	C6SVV1	60S ribosomal protein	protein. synthesis	−1.37	−0.40	1.35
38	C6SXD3	40S ribosomal protein S24	protein. synthesis	0.51	−0.52	−0.95
39	I1MDJ2	60S ribosomal protein L23A isoform	protein. synthesis	0.90	−0.67	−1.05
40	A0A0R0EIR6	glycosyltransferase	protein. synthesis	2.63	2.39	−1.93
41	I1N1W7	mitochondrial-processing peptidase subunit	protein. targeting	−0.41	−0.41	0.40
42	I1L0S5	outer envelope pore protein 16-2	protein. targeting	−0.93	−0.91	1.39
43	C6TB70	1-cys peroxiredoxin	redox	−0.36	−0.69	−0.41
44	I1KFE9	L-ascorbate oxidase	redox	0.62	1.10	0.42
45	I1LCG9	pre-mRNA-processing factor 17	RNA	−0.24	−1.45	0.86
46	K7LRU6	apoptotic chromatin condensation inducer	RNA	−0.79	−0.61	0.72
47	I1KEW2	O-methyltransferase 3	secondary metabolism	0.54	0.38	0.59
48	F8WRI3	gamma-tocopherol methyltransferase	secondary metabolism	1.06	1.37	−2.06
49	I1K711	uncharacterized protein	signalling	0.48	0.54	0.72
50	I1LBC8	mitochondrial Rho GTPase	signalling	−1.08	−1.10	0.93
51	K7KIL0	GTP-binding nuclear protein	signalling	1.30	−0.30	−0.80
52	K7LDT9	low-temperature-induced 65 kDa protein	stress	−1.86	−1.96	−1.01
53	C6T0C7	trypsin inhibitor A	stress	0.88	1.11	1.20
54	A0A0R4J4G9	heat shock 22 kDa protein	stress	−1.19	−2.53	−1.19
55	A0A0R4J4S6	NADPH-protochlorophyllide oxidoreductase	tetrapyrrole synthesis	0.60	−0.57	0.90
56	C6TDZ1	aquaporin PIP2-7	transport	−0.68	1.25	0.92
57	I1JMI9	malic enzyme	tricarboxylic acid cycle	0.55	0.20	−0.38
58	P15490	stem 28 kDa glycoprotein	miscellaneous	1.55	0.92	1.54
59	K7L6U4	uncharacterized protein	miscellaneous	0.74	0.56	0.82
60	P10743	stem 31 kDa glycoprotein	miscellaneous	1.59	0.77	0.85
61	I1KZI4	2,3-dimethylmalate lyase	not assigned	−0.87	0.98	1.17
62	I1MRP7	uncharacterized protein	not assigned	−0.67	−0.66	1.51
63	K7LKS0	exocyst complex component SEC5	not assigned	−0.49	−1.08	−1.26
64	I1LV40	uncharacterized protein	not assigned	−0.46	−0.76	−0.70
65	I1NA37	uncharacterized protein	not assigned	−1.11	−1.91	0.86
66	K7MGN4	trafficking protein particle complex subunit	not assigned	0.57	0.66	−0.49

^a^ “accession” is determined according to UniprotKB *Glycine max* (Soybean) protein database. ^b^ “functional category” is obtained using MapMan bin codes. Abbreviations: cell, cell division/organization; mitoETC, mitochondrial electron transport chain; protein, protein synthesis/targeting/others; RNA, RNA processing/regulation of transcription. “not assigned” indicates protein without ontology or characterized functions. ^c^ “fold change” indicates log_2_ fold change of identified proteins from millimeter waves irradiated soybean compared to control.

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
