# Peer review of "Proteomic Analysis of Irradiation with Millimeter Waves on Soybean Growth under Flooding Conditions"

_ijms, 2020, doi:10.3390/ijms21020486_

Round 1

Reviewer 1 Report

The manuscript by Zhong et al. “Proteomic Analysis to Reveal the Effect on Growth of 2 Soybean Seeds Irradiated with Millimetre Waves” presents the results of the study on the effects of millimetre waves on soybean seedling growth and proteome changes under flooding and control conditions. The authors identified major biochemical pathways affected by the millimetre wave treatment and some compounds majorly characterizing the plant response.

The work presented here is of potential great impact for soybean as the flooding represent an important stress to cope with during the crop cycle. The results are sound and large enough yet it present some issues that should be addressed in order to make the work more impactful.

A general concern is on how the analysis on seedling correlate on the adult plant performance, therefore I invite the authors to properly justify the choice of the developmental stage and plant organs analysed. Are this kind of test on small plants predictive of an adult plant performance? The final goal of research in crop production is higher productivity, therefore, in order to validate more your results I suggest modifying the discussion by making more meaningful correlation of your observations and the plant production potential, and to advance hypothesis on the plant productive/adaptive physiology instead of mainly presenting proteins and pathways that were increased or decreased. This would add more originality to the work and highlight the importance on the application of millimetre waves. Fundamentally, apart from the identification of differential protein content, the work fails in describing well the implication of results for the soybean growth and adaptability to stress. The amount of results presented is impressive yet it seems that the most important findings i.e. main differentially expressed proteins or altered pathways are not enough highlighted, the reader is often found with general descriptions of increased or decreased mechanisms. I suggest to include, when possible, more percentage or fold change comparison of the treated vs. controlled conditions. The way of representation of results in figures does not help in this, as there are many redundant information in the captions and actual figures (see the attached file for some details). Also, the understanding of and the knowledge on biological and biochemical concepts in plant response to stress seem to be considered known by the reader, which reduces the readability. The authors should provide more clarification throughout the text on why certain pathways/proteins have been analysed more in detail (see the attached file for details, e.g. lines 186, 210). As the flooding is a central argument of the manuscript I suggest changing the title in order to reflect better the results presented – mention “flooding conditions” in your title. The role and importance of trehalose should be introduced/mentioned in the Introduction Materials and methods section should be improved by adding the missing details (see the attached file) on sample size, replicas etc. There is also a redundancy in explaining a part of methodology in each figure/table caption. It is suggested that al the methodology is described only in the Materials and Method section and captions to contain a stand-alone description of given results. Consider using terminology such as “treated” and “control” instead of “irradiated” and “unirradiated” for better readability. Some specific and minor suggestions can be seen in the attached file.

Reviewer 2 Report

Unfortunately, I cannot recommend the acceptance of the manuscript at the current stage due to the following reasons.

Results presented in the manuscript are interesting and they will be interesting to the scientific community. But the manuscript is not coherent and needs improvements. At the beginning of the ‘Results’ section is described how the parameters of irradiation were experimentally selected, and there are no doubts that it was well done. Next, data are presented which were obtained from soybean seedlings cultivated under the following conditions:

A: unirradiated and unflooded

B: irradiated and unflooded

C: unirradiated and flooded

D: irradiated and flooded

The problem is that only Figures 2 and 3 concern all above mentioned experimental conditions. Figures 4 and 5 concern only A and B, but Figure 6 concerns only C and D. Figures 7 and 8 present results concerned A, B, C, and D, but the irradiation parameters are not the same as was applied in experiments which results are presented in Figures 2-6. In my opinion, all results must concern all four experimental conditions, so missing results (experimental variants) must be added to the manuscript. Additionally, only results coming from the same irradiation parameters must be presented (Figures 7 and 8).

It is not fully understood why the effect of trehalose was investigated. It is written that it was based on data presented in Figure 6 (line 256-257) but in my opinion Figure 6 is not enough for such a statement. Additionally, the parameters of irradiation in the experiments with the using of trehalose are not described. I suppose that sucrose or glucose also can improve the growth of soybean seedlings under flooding because these three sugars (glucose, sucrose, and trehalose) can regulate plant metabolism via changes in expression of many genes. It is visible in Figure 6 the level of hexokinase was clearly changed in the two presented experimental variants, so why glucose or sucrose was not taken under consideration. So, the authors must explain in detail why exactly the effect of trehalose was investigated, or better, this part of the results should be removed from the manuscript because it is not so important compared to the rest of the results.

Minor points

‘Flooding’ should be added to the title of the manuscript. Keywords should not be the same as words already used in a title. The sentence ‘Soybean seeds were irradiated without or with 5, 10, 20, and 40 mW of millimeter waves’ should be written in another way because irradiation without millimeter waves does not make sense. What does mean ‘at different stages of soybean’ in the heading of Table 1? It is not clear because only one time-point was investigated.

Reviewer 3 Report

This is an interesting and novel paper which greatly advances knowledge of the effects of millimeter-wavelength radiation in a model plant system.

Extensive spelling, grammar, and style edits are required here; the language is difficult to read.

Supplemental information was not included but is heavily used in this manuscript.

Why was proteomic analysis only performed on the root?  The shoot appears considerably larger as well.  This is particularly strange in light of photosynthesis-linked proteins having been identified as differentially regulated in the root.

More discussion of the role of trehalose and the link between trehalose and millimeter-wavelength irradiation would be useful, as this topic is introduced by is somewhat lacking in the discussion.  Why is trehalose useful in flood resistance?  How is this related to millimeter-wavelength irradiation?

Round 2

Reviewer 1 Report

Changes made in the manuscript (MS) by Zhong et al. are evident but not meaningful enough to increase the impact of the results. The authors have put effort to respond to the suggestions, yet some of the changes were made segmentally and not throughout the whole text, although comments previously given referred to the whole MS. With exception of minor changes to the text and synthesis of figures that increased readability, the results still remain a list of protein/pathways up- or down-regulated under the two conditions and do not emphasise in a straightforward way the potential of the millimetre wave treatment (for one, but not the only, example in the MS, see line 202 in the attached file). The amount of results in the main text and supplementary material is significant, and to all of them the same importance is given when commented, which often results complex to follow and grasp the key findings. The authors are invited to revise the data and carefully make a selection of those to include in the main text, as well as to revise the message they want to give to readers, besides the list of biochemical compounds changed under the studied treatment. The correlation between the results and possible implication on soybean growth and productivity was not integrated in a satisfactory way – the added paragraph 3.5. contains mainly the information that is suitable for the Introduction and does not merge well with the rest of the discussion. The Discussion section was overall enriched, but often with repetition of the results, and no proper development of the most important findings was given in relation to plant potential productivity see the attached file for some additional minor comments

Reviewer 2 Report

The manuscript is amended in accordance with my suggestions and requirements. The results of immunoblot analysis are the same as in the original submission, but the Authors explained enough why these results are not corrected. There are many other improvements in the manuscript, thus I recommend the acceptance of it. However, one new mistake must be corrected. Namely, the wording ‘in roots including hypocotyl’ (lines 98, 153, 240, 414, and 420) is incorrect. Such a phrase suggests that hypocotyl is a part of the root, but hypocotyl is a part of a shoot. Such wording is misleading and should be corrected. Maybe it should be simply written ‘in roots and hypocotyls’.
